# Proteomic Characterization of the Oral Pathogen *Filifactor alocis* Reveals Key Inter-Protein Interactions of Its RTX Toxin: FtxA

**DOI:** 10.3390/pathogens11050590

**Published:** 2022-05-17

**Authors:** Kai Bao, Rolf Claesson, Peter Gehrig, Jonas Grossmann, Jan Oscarsson, Georgios N. Belibasakis

**Affiliations:** 1Division of Oral Diseases, Department of Dental Medicine, Karolinska Institutet, 14104 Huddinge, Sweden; 2Division of Oral Microbiology, Department of Odontology, Umeå University, 90187 Umeå, Sweden; rolf.claesson@umu.se (R.C.); jan.oscarsson@umu.se (J.O.); 3Functional Genomics Center Zurich, ETH Zurich and University of Zurich, Winterthurerstrasse 190, 8057 Zurich, Switzerland; peter.gehrig@fgcz.ethz.ch (P.G.); jg@fgcz.ethz.ch (J.G.); 4SIB Swiss Institute of Bioinformatics, 1015 Lausanne, Switzerland

**Keywords:** *Filifactor alocis*, label-free quantification proteomics, FtxA

## Abstract

*Filifactor alocis* is a Gram-positive asaccharolytic, obligate anaerobic rod that has been isolated from a variety of oral infections including periodontitis, peri-implantitis, and odontogenic abscesses. As a newly emerging pathogen, its type strain has been investigated for pathogenic properties, yet little is known about its virulence variations among strains. We previously screened the whole genome of nine clinical oral isolates and a reference strain of *F. alocis*, and they expressed a novel RTX toxin, FtxA. In the present study, we aimed to use label-free quantification proteomics to characterize the full proteome of those ten *F. alocis* strains. A total of 872 proteins were quantified, and 97 among them were differentially expressed in FtxA-positive strains compared with the negative strains. In addition, 44 of these differentially expressed proteins formed 66 pairs of associations based on their predicted functions, which included clusters of proteins with DNA repair/mediated transformation and catalytic activity-related function, indicating different biosynthetic activities among strains. FtxA displayed specific interactions with another six intracellular proteins, forming a functional cluster that could discriminate between FtxA-producing and non-producing strains. Among them were FtxB and FtxD, predicted to be encoded by the same operon as FtxA. While revealing the broader qualitative and quantitative proteomic landscape of *F. alocis*, this study also sheds light on the deeper functional inter-relationships of FtxA, thus placing this RTX family member into context as a major virulence factor of this species.

## 1. Introduction

*Filifactor alocis* is a Gram-positive asaccharolytic, obligate anaerobe of the Firmicutes phylum that has recently been identified as a member of the oral microbiome with potential involvement in oral disease [1,2,3,4,5,6,7]. Its purported virulence mechanisms include the ability to manipulate neutrophils [8,9,10,11,12] and macrophages [13]. By whole-genome sequencing, we discovered that 60% of the *F. alocis* strains encode a novel repeats-in-toxins (RTX) protein family member, which we designated as FtxA [14]. Several of the virulence factors and/or putative virulence-related proteins of *F. alocis*, including FtxA, are contained within extracellular vesicles released by this organism [15,16]. Further to whole-genome sequencing, protein variations in the presence and expression levels among *F. alocis* strains can be expected. Hence, the early genomic characterization of *F. alocis* needs to be supported by universal quantitative proteomics, which will enable the deeper characterization of intraspecies proteinic differences and identification of relevant virulence-associated motifs. Earlier proteomic work identified protein differences (i.e., in cell-wall anchoring proteins) among two *F. alocis* strains, which may reflect variations in virulence between them, evidenced by their differential effects on host cells [17]. By using a quantitative shotgun proteomics platform and an in-house proteomics database, the present study aimed at defining the full proteomes of the ten *F. alocis* strains (nine clinical isolates and one reference strain) to identify differences among their core proteomes and protein–protein interaction patterns. Furthermore, specific focus was placed on the functional interactions of FtxA with other proteins as well as the differential abundance regulation of FtxA between strains.

## 2. Results

### 2.1. Proteome Profiles of Ten *F. alocis* Strains

To analyze the full proteome differences between *F. alocis* strains, a total of 872 proteins were identified and quantified from the reference strain ATCC 35896 and nine *F. alocis* clinical isolates (*n* = 3 for each strain) (Appendix A) based on our Progenesis QI-Scaffold inclusion criteria. Among them, 744–802 proteins were identified from each strain (Table 1). The visual representations of protein abundances and the correlation between strains are provided as heatmaps in Figure 1. Unsupervised clustering of the heatmap data revealed that the biological replicates within the same strain (e.g., 10E-17U_1, 10E-17U_2, and 10E-17U_3 were triplicates of strain 10E-17U) were grouped separately from others, except for 624B-08U_1, indicating that almost all strains were indeed distinctive at the proteomic level. In addition, the clustering results also showed that protein profiles of 624B-08U were the closest to the reference strain ATCC 35896 among all nine clinical isolates, while the profiles of strains 6B-17U and 413B3-17U were the most distinctive. We previously found a putative member of the large repeats-in-toxins (RTX) toxin family, FtxA, which is consistent with phylogenetic relationships based on multilocus sequence typing analysis [14]. In line with this discovery, both 6B-17U and 413B3-17U do not express FtxA.

### 2.2. Differential Expression of Proteins between FtxA-Positive and -Negative Strains and Their Predicted Functional Protein Association Networks

Thirty-four proteins were differentially expressed (abs (Log2FC) > 1 and *p*-value < 0.05) in *F. alocis* strains with *ftxA*-positive genotypes compared with *ftxA*-negative strains (Table 2 and Appendix A). Yet, 41 proteins were exclusively identified in *ftxA*-positive strains, while seven were exclusively identified in *ftxA*-negative strains (Table 3 and Appendix A). In addition, 12 proteins were found to be at least twice as high in one condition than the other but unable to have a *p*-value, since only one sample was identified in their weaker conditions (Table 4 and Appendix A). To further understand their function, the inter-relationships of all differentially expressed proteins were investigated by STRING. In sum, 66 pairs of functional associations with a combined confidence score >0.4 were retrieved among 44 (including FtxA(UniProt ID ADW1614)) differentially expressed proteins (Figure 2 and Appendix A). Although most of these associations only involved two or three proteins, probably due to the lack of known information on *F. alocis*, three of them involved multiple proteins and formed three protein network clusters (Figure 2). The largest clustering includes five proteins having at least some part of their peptide sequence embedded in the hydrophobic region of the membrane (i.e., integral component of membrane Gene Ontology Term (GO:0016021)). Yet, their annotated functions are quite distinctive including DNA repair (EFE28003.1), DNA-mediated transformation (EFE28216.1), type II secretion system (EFE28506.2), and pilin domain protein (EFE28505.1). The second-largest cluster of these three, with five different proteins, were mainly proteins with catalytic activity, except EFE28863.1, which is an ABC transporter. The smallest cluster of these three was a group of enzymes including dehydrogenase and aminotransferase.

### 2.3. Predicted Functional Protein Association Network for FtxA

Additional protein interaction analysis was centered on the novel RTX family member of *F. alocis*, FtxA. The STRING protein–protein interaction analysis revealed that six proteins had interactions with FtxA (ADW16141.1), thereby forming a putative “functional FtxA cluster” of seven proteins (Figure 3A and Appendix A). This included three proteins from the *ftx* gene cluster itself: FtxA (ADW16141.1), FtxB (EFE27661.1), and FtxD (EFE27662.1), as well as four other essentially uncharacterized proteins. Four of these seven proteins, including FtxA, were identified and quantified in this work (Figure 3B). The identification of ADW16141 (FtxA) was consistent with the *ftxA* genotypes in our previous work [14]. Of the remaining three identified and quantitated proteins, one was annotated as a “repeat protein” and contained a copper amine oxidase N-terminal domain with a divergent InlB B-repeat domain (ADW16149.1). This protein displayed interactions with only two of the seven cluster proteins, apart from FtxA (Figure 3A). The uncharacterized protein EFE27658.1 was encoded directly upstream of *ftxA*, whereas another uncharacterized protein EFE27629.1 was also found to interact with FtxA, mainly in automated text mining and other annotations from STRING (Appendix A). However, there is no clear functional overlap between these proteins based on their predicted functions (Table 5). Of note, these predicted proteins are still in the early stage of annotation. As a result, none of them have an assigned function in the KEGG database, and no BRITE terms have been generated.

## 3. Discussion

In this study, we analyzed the full proteomes of ten *F. alocis* strains, which yielded a total of 872 proteins, the majority of which were identified in all strains. For instance, 802 proteins were identified in the ATCC 35896 strain, 755 proteins were identified in the 845G-16U strain, and 762 proteins were identified in the 117A-17U strain. In comparison to the reference strain, ATCC 35896, which is so far the best-characterized one, strain 624B-08U showed the closest proteomic profile identity, whereas strains 6B-17U and 413B-17U were the most distant in this respect. This is in agreement with the phylogenetic relationships revealed among the ten *F. alocis* strains, based on eight genes using multilocus sequence typing analysis [14]. Of note, those two strains were isolated from different infectious sources, as the former was a constituent of dental biofilm at a periodontitis site, whereas the latter at an acute necrotizing gingivitis (ANUG) site. Hence, the specialized ecological niche of the infection could account for qualitative and quantitative proteomic variations among clinical isolates. We observed that strains expressing FtxA, a putative member of the large repeats-in-toxins (RTX) toxin family [14], revealed more common virulence characteristics, regardless of their infectious origin, and possibly associated with activities on host immunity [18]. Clustering of bacterial strains according to the expression levels of RTX toxins has been observed for other species, including *Escherichia coli* [19,20], and the periodontal pathogen *Aggregatibacter actinomycetemcomitans* [21,22]. Clustering has also been seen based on other toxins, such as for *Salmonella enterica* serovar Typhimurium strains expressing cytolethal distending toxin (CDT) and other serovar Typhi-related genes [23].

Ninety-seven proteins were differentially regulated in FtxA positive compared with FtxA negative groups. While the differential abundances of proteins may account for variations in functional and biosynthetic activities between strains, they merely imply differences in virulence characteristics. Indeed, both strains were isolated from infected root canals, even though they largely differed in terms of expressed protein abundances with only one expressing FtxA. We also attempted to evaluate the functions of all differentially expressed proteins, using enrichment analysis based on known databases, that did not have significant results (data not shown). Despite the fact that the genome of *F. alocis* has been annotated, most of their proteins were only computationally analyzed (i.e., unreviewed proteins), and their automatic annotation functions were sometimes not sufficient for accurate enrichment analysis. Alternatively, the largest clustering from protein associate networks constitutes a superfamily of integral membrane proteins that mediate ATP-powered translocation of many substrates across membranes, either for import or export [24]. Proteins clustering with catalytic activity or clustering of various dehydrogenase and aminotransferase were also found by String. The contribution of ABC transporters [25], respective to antibiotic resistance in many bacterial species, was demonstrated and was in agreement with the few ABC transporters we discovered in this work. Hence, based on our analyses, a key variation among the different *F. alocis* strains might rely on their antibiotic resistance capabilities, which, however, need to be validated in further studies.

Finally, we considered the associations between the expressions of FtxA protein and other proteins associated with it. We observed that the presence and absence abundance-based of FtxA was consistent with the *ftxA* genotypes in our previous work [14], which is a good indicator that we applied a reliable protein-inclusion criteria in the work. The STRING protein–protein interaction analysis revealed that six other proteins can interact with FtxA and, hence, tentatively constitute a cluster of proteins that may be functionally associated with cytoplasmic FtxA, three of which (in addition to FtxA) were identified and quantitated in the present work. The *ftx ABD* gene operon encoded four predicted products [14], hypothetical protein EFE27658.1, FTXA, FtxB, and FtxD; the last two were not identified in the current study. Whether the hypothetical protein EFE27658.1, encoded directly upstream of *ftxA*, has any role in FtxA post-translational modification, intracellular trafficking, and/or secretion is not known. In addition, it displays no apparent similarity to an equivalent, such as HlyC or TolC, commonly present in and/or associated with RTX toxin-encoding gene clusters [14,19]. The other two proteins can potentially interact with FtxA thanks to their proximity within the chromosome where they are encoded (i.e., chromosome neighborhoods) based on the neighborhood prediction algorithm of STRING as well as other annotations. However, we should also beware that these two proteins were not encoded in an operon with FtxA. The FtxA-associated “repeat protein” ADW16149.1 appeared to be present in all strains, including the strains lacking *ftx*A, and it had no apparent sequence similarity to *ftx*A, neither was it encoded close to the *ftx* gene cluster. Interestingly, however, this FtxA-associated protein appears to be an InlB B-repeat-containing protein, which may associate it with host cell invasion [26]. This remains to be experimentally tested. Since there is currently no clear overlap based on their predicted functions, these proteins are still in the early stage of discovery and, thus, warrant deeper exploration.

In conclusion, the present study identified that *F. alocis* species has a broad “core proteome”, while there are also quantitative variations in the expression of select proteins between strains. The functional pathways associated with the most or least abundantly expressed proteins were related to ribosomal and mitochondrial activity as well as protein biosynthesis and transportation. Due to the early stage of bioinformatic annotation of the identified proteins, it is difficult to confer any deeper roles in the metabolic functions of this species, let alone in the virulence-specific characteristics of individual strains. Nevertheless, the global proteomic analysis of *F. alocis* performed in this study justifies the need for a deeper characterization of its recently discovered FtxA RTX toxin. Indeed, our analysis revealed that a functional clustering of specific protein–protein interactions can discriminate between FtxA-producing and non-producing strains. The identities, functions, and interactions of these proteinic groups need to be further investigated to reveal whether they comprise a pathogenicity island *within F. alocis* that could regulate the virulence of this species.

## 4. Materials and Methods

### 4.1. *F. alocis* Strains and Growth Conditions

The *F. alocis* reference strain ATCC 35896 (also known as CCUG 47790) was purchased from the Cultural Collection of the University of Gothenburg (CCUG) [27,28]. The nine *F. alocis* clinical isolates used in the present study (i.e., 854G-16U, 117A-17U, 149A-17U, 624B-08U, 373F-17U, 6B-17U, 10E-17U, 413B3-17U, and 148B-17U) were isolated and characterized at the Clinical Microbiology Laboratory, Umeå University, as described previously [14]. Fastidious anaerobe agar (FAA; Neogen^®^, Heywood, UK) plates were used to culture all strains in an anaerobic environment (i.e., 10% H_2_, 5% CO_2_, and 85% N_2_) at 37 °C.

### 4.2. Bacterial Protein Extraction

The *F. alocis* strains were cultivated for three days under the condition described above before being suspended in PBS. The *F. alocis* suspensions were then adjusted to approximately OD600 nm = 1.0. Then, 0.5 mL of suspensions in each strain were reduced, alkylated, trypsinized, and purified using the PreOmics iST kit (PreOmics GmbH) following the manufacturing protocol for protein extraction and digestion. These extracts were concentrated using a Speedvac (Thermo Savant SPD121P, Thermo Scientific, Waltham, MA, USA) and stored at −20 °C until further use.

### 4.3. LC-MS/MS Analysis

The bacterial extracts were first reconstituted with 30 µL of 3% acetonitrile (ACN) in 0.1% formic acid, then normalized to 1 mg/mL based on the estimated protein concentration using a NanoDrop One system (Thermo Fisher Scientific, Madison, WI, USA). One microgram of each sample was then loaded on an Orbitrap Fusion Tribrid mass spectrometer (Thermo Fisher Scientific, San Jose, CA, USA) interfaced to an Easy nano-flow HPLC system (Thermo Fisher Scientific) in a randomized order for mass spectrometry analysis. A pool of all samples was inserted around the middle of the sequence to be used as the reference for the label-free quantification. The liquid chromatography solvent compositions of buffers A and B were 0.1% formic acid in water and 0.1% formic acid in acetonitrile, respectively. The samples were loaded onto an Acclaim PepMap 100 (Thermo Scientific) trap column, 75 μm × 2 cm, packed with C18 material, 3 μm, 100 Å, and separated on an analytical EASY-Spray column (Thermo Scientific, 75 μm × 500 mm) packed with reverse-phase C18 material (PepMap RSLC, 2 μm, 100 Å). Peptides were eluted over 110 min at a flow rate of 300 nL/min. The following LC gradient protocol was applied: 0–2 min: 2% buffer B; 95 min: 25% B; 100 min: 35% B; 105–110 min: 95% B.

Survey scans were acquired in the Orbitrap mass analyzer in the range of m/z 300–2000, with a resolution of 120,000, an automated gain control (AGC) target value of 400,000, and a maximum injection time of 50 ms. Higher energy collisional dissociation (HCD) spectra were acquired in the linear ion trap mass analyzer, using a normalized collision energy of 30%. Precursor ions were isolated in the quadrupole with an m/z 1.6 isolation window. Charge state screening was enabled, and only precursor ions with charge states of 2–7 were included. The threshold for signal intensities was 5000. Precursor ion masses already selected for MS/MS acquisition were dynamically excluded for 25 s. A maximum injection time of 300 ms, an AGC target value of 2000, and a first mass of 140 for HCD spectra were applied.

### 4.4. Label-Free Quantification

Label-free quantification was performed using Progenesis QI for Proteomics (Nonlinear Dynamics) as described previously [29]. In brief, all raw files were aligned with the pooled sample for feature detection, alignment, and quantification. An mgf file of all aligned samples was exported with the top 5 ms/ms per feature, 200 minimal fragment ion count, and deisotoping and charge deconvolution, and then it was exported for searching using Mascot (version 2.4.1, Matrix Science, London, UK) against an in-house database containing 6679 protein sequences. This database included *F. alocis* proteins (taxon identifiers 143361 and 546269, downloaded from UniProt (https://www.uniprot.org/) (accessed on 19 March 2018), 260 sequences, known as MS contaminants, and reverse sequences were used as a decoy for estimating the false discovery rate (FDR) [30]. The following search parameters were used: precursor tolerance: ±10 ppm; fragment ion tolerance: ±0.6 Da; enzyme: trypsin; maximum missed cleavages: 2; fixed modification: carbamidomethyl (C); variable modification: oxidation (M) and acetyl (protein N-term). Then, the spectrum reports of the search result were generated using Scaffold (version 4.2.1, Proteome software) with a threshold of protFDR of 10%, minimum of one peptide, and a pepFDR of 5%, which was imported in Progenesis QI for Proteomics for identifying the quantified proteins.

To minimize potential errors introduced by aggressively matching features between samples in Progenesis QI for Proteomics. All raw files from each sample were also individually searched using Mascot against the same database, with the following searching parameters: precursor tolerance: ±10 ppm; fragment ion tolerance: ±0.6 Da; enzyme: trypsin; maximum missed cleavages: 2; fixed medication: carbamidomethyl (C); oxidation (M) and acetyl (protein N-term). These Mascot generic files (mgf) were combined using Scaffold (version 4.2.1, Proteome software, Portland, OR, USA) and then exported using Scaffold at a cutoff at 3.0% FDR at the protein level (protFDR), minimal two peptides, and 1.0% FDR at the peptide level (pepFDR). Then Progenesis results were compared with the Mascot results. The Progenesis quantified proteins were only accepted as ture quantifications if they were also identified from an individual mgf in a Mascot search with a minimum of 2 unique peptides. The normalized abundances from these accepted proteins were then kept for quantification, while the abundances from proteins not identified in individual mgfs were replaced with “NA”. These Mascot-filtered Progenesis results were used to calculate fold changes (FCs) between strains in the FtxA-positive compared with the FtxA-negative group as well as log2 transformed FC. The hyperbolic arcsine transformed result was used for two-tailed student *t*-tests as in Progenesis QI. Proteins with an absolute value of log2FC > 1 as well as a *p*-value < 0.05 were considered as being regulated. Benjamini–Hochberg FDR corrections were provided based on the *p*-value.

Some proteins were identified and quantified in either *ftxA*-positive or -negative strains (i.e., only found in one condition). Therefore, they cannot have FC or *p*-values. Similarly, other proteins that were found to display high abundance changes (absolute value of log2FC > 1) between *ftxA*-positive or -negative strains cannot acquire *p*-values due to they have only one identification in one of the conditions. Nevertheless, proteins with high intensity in one condition but not present in the other condition (or present in a low abundance) can have biological relevance. Thus, proteins in the above two circumstances were also defined as regulated proteins.

All three types of regulated proteins, namely, (a) proteins differentially expressed (abs (Log2FC) > 1 and *p*-value < 0.05), (b) proteins exclusively identified in one condition, and (c) proteins expressed at least twice as high in one condition than the other (abs (Log2FC) > 1) but with no *p*-value (could not acquire a *p*-value, as only one sample was identified in one of the conditions) were treated equally in the following functional analysis.

### 4.5. Data Clustering and Heat Maps for Regulated Proteins

The R software (R: A Language and Environment for Statistical Computing, R Development Core Team) and, in particular, the packages quantable (https://cran.r-project.org/web/packages/quantable/index.html) (accessed on 11 September 2019) and pheatmap (https://cran.r-project.org/web/packages/pheatmap/index.html) (accessed on 24 October 2019) were used to generate unsupervised clustering analysis, correlations between different strains and heat maps. No apparent outlier was found or excluded in this study.

### 4.6. Functional Analysis for Regulated Proteins

The enrichment analyses were conducted in the STRINGdb package, version 3.1.3, on 20 July 2021, using all quantified proteins as background. The interaction scores were calculated from experimental evidence as well as predictions based on knowledge gained from other organisms [29], using STRING (https://string-db.org/) (accessed on 1 February 2021). Only proteins with medium confident scores (>0.4) are shown in the illustration. The function of proteins that contributed to the enriched functions or pathways were manually searched in KEGG (https://www.genome.jp/kegg/) (accessed on 15 September 2021) to retrieve their BRITE hierarchical classifications and Pfam domain annotations.

### 4.7. Predicted Interaction for FtxA

The protein–protein interactions predicted for FtxA were determined using STRING (https://string-db.org/) (accessed on 11 August 2021). All seven independent channels for STRING interaction analysis including, chromosome neighborhoods, gene fusion, phylogenetic co-occurrence, homology co-expression, experimentally determined interaction, database annotation, and automated text mining, were used to identify interactions. Proteins that exhibited a final combination score with a medium confidence of more than 0.4 were considered.

### 4.8. Image Processing

Microsoft PowerPoint (version 16; Microsoft, Redmond, WA, USA) was used for assembling the figures.

### 4.9. Ethical Considerations

All procedures were conducted following the guidelines of the local ethics committee at the Medical Faculty of Umeå University, which are based on the Declaration of Helsinki (64th WMA General Assembly, Fortaleza, October 2013).

### 4.10. Data Availability

Mass spectrometry data were handled using the local laboratory information management system (LIMS) [31]. The in-house database and mass spectrometry proteomics data were deposited to the ProteomeXchange Consortium via the PRIDE partner repository with the dataset identifier PXD026971. The raw file names and their corresponding sample names are listed in Appendix A. The authors declare that all data supporting the findings of this study are available within the article and the Appendix A or upon request from the corresponding author.

## Figures and Tables

**Figure 1 pathogens-11-00590-f001:**
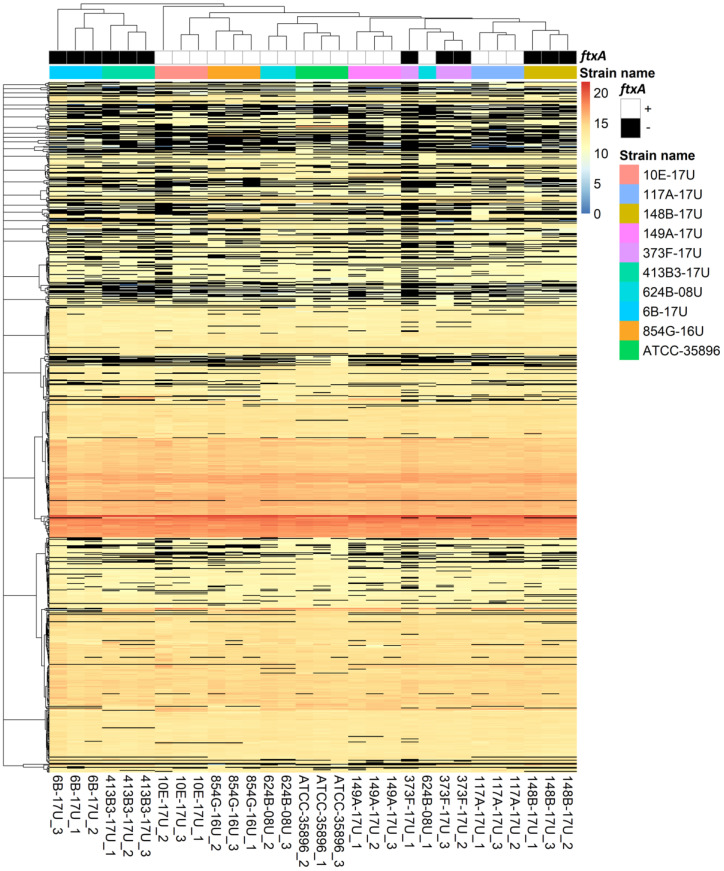
Heat map of the normalized abundance for identified and quantified proteins. The colors in the map display the value of the arcsinh transformed normalized abundance value plus one for individual proteins (represented by a single row) within each experimental sample (represented by a single column). Expression values are shown as a color scale, with higher values represented by red and lower represented by blue. The normalized abundance values of “NA” are represented by black. The *ftxA* genotypes (+ or −) [14] and different strains are color coded.

**Figure 2 pathogens-11-00590-f002:**
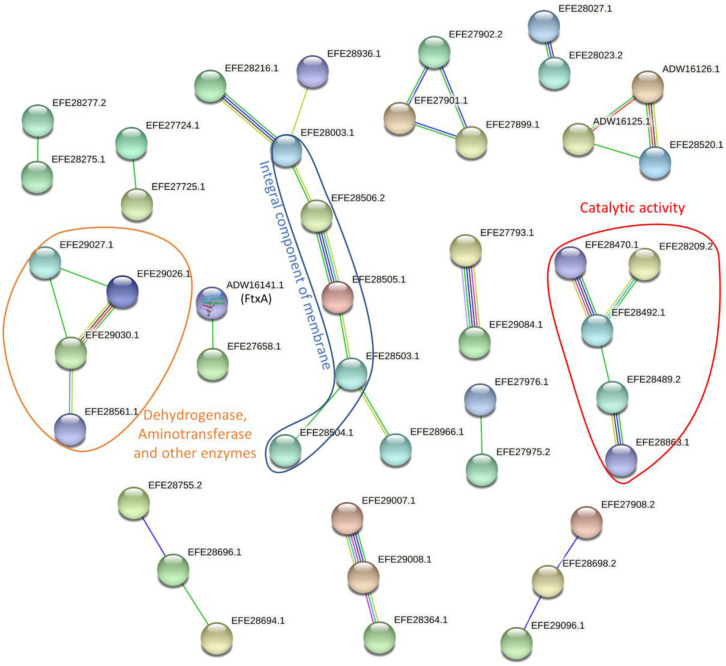
Protein–protein interactions between regulated proteins. The network established using STRING 10.5 showed protein–protein interactions with a medium confidence score (0.4) (Appendix A). The colors of the lines illustrate different types of interactions. Among them, the blue and purple lines indicate interactions based on the curated database and experimental results, respectively, while green, red, dark blue, yellow, and black lines are predicted interactions determined from gene neighborhood, gene fusions, gene co-occurrence, text mining, and co-expression, respectively.

**Figure 3 pathogens-11-00590-f003:**
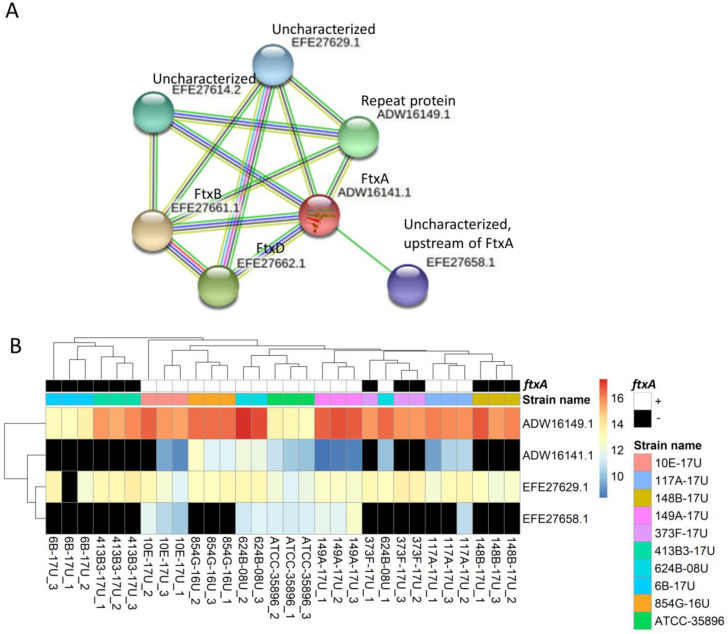
Predicted protein interactions of FtxA and their detected expressions of FtxA and interacting proteins in the different strains. (**A**) The network established using STRING 10.5 showed protein–protein interactions with a medium confidence score (0.4) (Appendix A). The colors of the lines illustrate different types of interactions as is shown in Figure 2. Four identified proteins are highlighted in circles. (**B**) The abundance of identified proteins is displayed in the values for arcsinh transformed normalized abundance plus one in the heatmap. The normalized abundance values of “NA” are represented by black. The clustering between rows is based on four identified proteins, while the clustering between columns is based on all identified proteins (same as Figure 1).

**Table 1 pathogens-11-00590-t001:** The number of identified and quantified proteins from each strain based on Progenesis QI-Scaffold inclusion criteria.

Strain Name	ATCC 35896	854G-16U	117A-17U	149A-17U	624B-08U	10E-17U	373F-17U	6B-17U	413B3-17U	148B-17U
No. of proteins	802	755	762	761	785	744	746	762	746	762

**Table 2 pathogens-11-00590-t002:** Differentially expressed proteins (i.e., abs (log2FC) > 1, *p* < 0.05).

No.	First Accession	Unique Peptides	log2FC FtxA+/FtxA−	*t*-Test	Description
1	EFE27669.1	3	−6.98	5 × 10^−^^13^	hypothetical protein HMPREF0389_01588
2	EFE28470.1	4	−4.23	7 × 10^−^^3^	DNA polymerase III, alpha subunit
3	EFE28966.1	9	−2.57	3 × 10^−^^2^	ABC transporter, substrate-binding protein, family 3
4	EFE28389.1	12	−2.28	3 × 10^−^^2^	UDP-N-acetylmuramoyl-tripeptide--D-alanyl-D-alanine ligase
5	EFE29056.2	4	−1.78	3 × 10^−^^2^	hypothetical protein HMPREF0389_00978
6	EFE29171.2	4	−1.65	4 × 10^−^^2^	HIRAN domain protein
7	EFE29153.1	9	−1.58	3 × 10^−^^2^	transcriptional regulator, Spx/MgsR family
8	WP_083799680.1	2	−1.43	6 × 10^−^^3^	aromatic acid exporter family protein
9	EFE28489.2	13	−1.31	9 × 10^−^^4^	GTP-binding protein TypA
10	EFE28520.1	2	−1.30	2 × 10^−^^2^	Sua5/YciO/YrdC/YwlC family protein
11	EFE27675.1	10	−1.18	3 × 10^−^^3^	RelA/SpoT domain protein
12	EFE28371.1	4	−1.11	3 × 10^−^^3^	aminotransferase, class V
13	EFE28992.1	6	−1.03	4 × 10^−^^4^	B3/4 domain protein
14	EFE28003.1	3	−1.02	3 × 10^−^^4^	comEA protein
15	EFE28083.2	27	1.02	6 × 10^−^^3^	LPXTG-motif cell-wall anchor domain protein
16	EFE28698.2	6	1.06	4 × 10^−2^	hypothetical protein HMPREF0389_00616
17	EFE28364.1	25	1.07	7 × 10^−^^8^	endopeptidase La
18	EFE27626.1	13	1.08	4 × 10^−^^2^	signal recognition particle-docking protein FtsY
19	EFE28863.1	7	1.11	8 × 10^−^^3^	ABC transporter, ATP-binding protein
20	EFE27917.1	7	1.28	1 × 10^−^^2^	CRISPR-associated protein, Csd1 family
21	EFE28620.1	7	1.68	5 × 10^−^^3^	transcriptional regulator, effector binding domain protein
22	EFE27793.1	61	1.70	6 × 10^−^^5^	LPXTG-motif cell-wall anchor domain protein
23	EFE28561.1	4	1.77	3 × 10^−^^2^	aspartate–ammonia ligase
24	EFE28977.1	2	1.81	3 × 10^−^^2^	cobalt transport protein
25	EFE29084.1	30	2.18	3 × 10^−^^3^	LPXTG-motif cell-wall anchor domain protein
26	EFE27653.2	3	2.40	3 × 10^−^^3^	hypothetical protein HMPREF0389_01572
27	EFE27796.2	3	2.40	3 × 10^−^^2^	hypothetical protein HMPREF0389_01427
28	EFE29085.1	11	2.41	2 × 10^−^^2^	purine nucleoside phosphorylase I, inosine and guanosine-specific
29	EFE28936.1	20	2.80	9 × 10^−^^4^	pyridoxal 5’-phosphate synthase, synthase subunit Pdx1
30	EFE28939.1	10	3.60	4 × 10^−^^2^	thermonuclease
31	EFE29027.1	28	5.05	3 × 10^−^^7^	TRAP transporter solute receptor, DctP family
32	EFE29030.1	27	5.44	8 × 10^−^^6^	aminotransferase, class I/II
33	EFE29026.1	22	8.84	2 × 10^−^^6^	4-phosphoerythronate dehydrogenase
34	EFE29077.1	2	12.39	4 × 10^−^^3^	response regulator receiver domain protein

**Table 3 pathogens-11-00590-t003:** Proteins found in only one condition.

No.	First Accession	Unique Peptides	Discovered in FtxA+ or FtxA− Strains	Description
1	ADW16126.1	2	FtxA+	protein-(glutamine-N5) methyltransferase, release factor-specific
2	ADW16141.1	9	FtxA+	type I secretion target GGXGXDXXX repeat (2 copies)
3	ADW16178.1	2	FtxA+	N-acetylmuramoyl-L-alanine amidase
4	EFE27600.1	2	FtxA+	hypothetical protein HMPREF0389_01518
5	EFE27658.1	4	FtxA+	hypothetical protein HMPREF0389_01577
6	EFE27681.2	53	FtxA+	poly(R)-hydroxyalkanoic acid synthase, class III, PhaE subunit
7	EFE27724.1	2	FtxA+	amino acid carrier protein
8	EFE27725.1	26	FtxA+	peptidase, M20/M25/M40 family
9	EFE27785.1	2	FtxA+	DEAD/DEAH box helicase
10	EFE27899.1	2	FtxA+	hypothetical protein HMPREF0389_01151
11	EFE27901.1	6	FtxA+	hypothetical protein HMPREF0389_01153
12	EFE27902.2	5	FtxA+	hypothetical protein HMPREF0389_01154
13	EFE27946.1	2	FtxA+	peptidase family T4
14	EFE27975.2	13	FtxA+	hypothetical protein HMPREF0389_01227
15	EFE27976.1	2	FtxA+	hypothetical protein HMPREF0389_01228
16	EFE27979.1	2	FtxA+	IclR helix-turn-helix domain protein
17	EFE28064.1	2	FtxA+	hypothetical protein HMPREF0389_01317
18	EFE28209.2	4	FtxA+	MutS2 family protein
19	EFE28216.1	2	FtxA+	DNA protecting protein DprA
20	EFE28277.2	9	FtxA+	hypothetical protein HMPREF0389_00192
21	EFE28328.2	3	FtxA+	addiction module antitoxin, RelB/DinJ family
22	EFE28411.1	2	FtxA+	hypothetical protein HMPREF0389_00326
23	EFE28456.1	2	FtxA+	ACT domain protein
24	EFE28503.1	19	FtxA+	prepilin-type cleavage/methylation N-terminal domain protein
25	EFE28504.1	7	FtxA+	hypothetical protein HMPREF0389_00419
26	EFE28505.1	18	FtxA+	prepilin-type cleavage/methylation N-terminal domain protein
27	EFE28506.2	8	FtxA+	bacterial type II secretion system domain protein F
28	EFE28572.1	3	FtxA+	transcriptional regulator, Fur family
29	EFE28693.1	7	FtxA+	hypothetical protein HMPREF0389_00610
30	EFE28694.1	4	FtxA+	hypothetical protein HMPREF0389_00612
31	EFE28696.1	3	FtxA+	hypothetical protein HMPREF0389_00614
32	EFE28755.2	3	FtxA+	hypothetical protein HMPREF0389_00675
33	EFE28756.1	11	FtxA+	alcohol dehydrogenase, iron-dependent
34	EFE28838.1	2	FtxA+	DNA-binding helix-turn-helix protein
35	EFE29023.1	4	FtxA+	hypothetical protein HMPREF0389_00945
36	EFE29055.1	2	FtxA+	DNA-binding helix-turn-helix protein
37	EFE29096.1	2	FtxA+	hypothetical protein HMPREF0389_01018
38	EFE29109.1	3	FtxA+	hypothetical protein HMPREF0389_01031
39	EFE29118.2	4	FtxA+	type I restriction modification DNA specificity domain protein
40	EFE29197.1	2	FtxA+	hypothetical protein HMPREF0389_01122
41	WP_041250771.1	7	FtxA+	acyl carrier protein
42	EFE28023.2	3	FtxA−	iron permease FTR1 family
43	EFE29170.2	2	FtxA−	hypothetical protein HMPREF0389_01093
44	EFE28866.1	2	FtxA−	dihydroorotate dehydrogenase, electron transfer subunit
45	EFE28492.1	4	FtxA−	excinuclease ABC, B subunit
46	ADW16125.1	2	FtxA−	hypothetical protein HMPREF0389_01679
47	EFE27676.1	2	FtxA−	VanW-like protein
48	EFE27908.2	2	FtxA−	hypothetical protein HMPREF0389_01160

**Table 4 pathogens-11-00590-t004:** Proteins unable to provide a *p*-value, but abs (Log2FC) > 1.

No.	First Accession	Unique Peptides	log2FC FtxA+/FtxA−	Description
1	EFE27582.2	2	1.32	hypothetical protein HMPREF0389_01655
2	EFE27612.1	4	1.40	phospholipase D domain protein
3	EFE29008.1	8	1.88	3-oxoacyl-[acyl-carrier-protein] reductase
4	EFE27694.1	2	2.31	adenosylmethionine-8-amino-7-oxononanoate transaminase
5	EFE29007.1	5	3.73	beta-ketoacyl-acyl-carrier-protein synthase II
6	EFE28275.1	9	3.75	DNA topoisomerase
7	EFE28848.1	12	4.87	hypothetical protein HMPREF0389_00770
8	EFE27992.2	2	5.71	EDD domain protein, DegV family
9	EFE28027.1	2	−3.13	efflux ABC transporter, permease protein
10	EFE27594.2	2	−3.05	hypothetical protein HMPREF0389_01512
11	EFE28953.1	2	−2.47	hypothetical protein HMPREF0389_00875
12	EFE28704.2	2	−2.23	hypothetical protein HMPREF0389_00623

**Table 5 pathogens-11-00590-t005:** The predicted functions for FtxA and the identified FtxA-related proteins.

Accession	Protein Domain (Pfam) *
ADW16141.1	RTX calcium-binding nonapeptide repeat (4 copies); Hemolysin-type calcium binding protein related domain; RTX C-terminal domain
ADW16149.1	Copper amine oxidase N-terminal domain; Listeria–Bacteroides repeat domain (List_Bact_rpt); Divergent InlB B-repeat domain; Family of unknown function (DUF6273)
EFE27629.1	Outer membrane efflux protein; SMODS- and SLOG-associating 2TM vector domain 1; Hydrogenase/urease nickel incorporation, metallochaperone
EFE27658.1	N/A

* Pfam is a database (http://pfam.xfam.org/) (accessed on 15 September 2021) of protein domain families. The definition of each Pfam term for the corresponding proteins is reported in the table. Terms are separated by semicolons.

## Data Availability

The in-house database and mass spectrometry proteomics data were deposited to the ProteomeXchange Consortium via the PRIDE partner repository with the dataset identifier PXD026971. The raw file names and their corresponding sample names are listed in Appendix A. The authors declare that all data supporting the findings of this study are available within the article and Appendix A or upon request from the corresponding author.

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
