# Peer review of "Proteomic Characterization of the Oral Pathogen Filifactor alocis Reveals Key Inter-Protein Interactions of Its RTX Toxin: FtxA"

_pathogens, 2022, doi:10.3390/pathogens11050590_

Round 1

Reviewer 1 Report

This study from Georgios Belibasakis group provides valuable information related to the similarities and differences among the F. alocis clinical strains and is centered around the presence or absence of the recently identified FtxA by the same group. Characterization of the potential virulence factors of F. alocis is valuable information to help delineate the pathogenic persona of the emerging oral pathogen and how it could contribute to periodontal disease pathogenesis. 

Minor comments

  • Will be of value to add a table with the 41 proteins that were exclusively present in the ftxA positive strains and the 7 ones that were only identified in the ftxA negative strains.
  • Line 94 has Figure 3E but there is no such figure in the manuscript. Please correct to indicate the appropriate figure.
  • Line 127: the way the sentence is written indicates that Table 1 shows predicted protein functions, but the table only shows the total number of proteins identified, not their function. Modify the text accordingly to reflect the table content. Unless the text was referring to Table 4?
  • Italicized of F. alocis in several parts of the discussion is missing. For example, lines 172, 182.

Author Response

Thank you for the time and efforts in reviewing the present manuscript and for giving these valuable and relevant comments. We have given the responses to the comments below and made changes in the manuscript accordingly and highlighted all amendments, in the revised manuscript.

Comments from Reviewer 1:

This study from Georgios Belibasakis group provides valuable information related to the similarities and differences among the F. alocis clinical strains and is centered around the presence or absence of the recently identified FtxA by the same group. Characterization of the potential virulence factors of F. alocis is valuable information to help delineate the pathogenic persona of the emerging oral pathogen and how it could contribute to periodontal disease pathogenesis. 

Minor comments

  • Will be of value to add a table with the 41 proteins that were exclusively present in the ftxA positive strains and the 7 ones that were only identified in the ftxA negative strains.

Response: We appreciated the Reviewer for this constructive feedback. A new table (table 2B) with proteins that are unique to the ftxA positive or negative strains was created.

  • Line 94 has Figure 3E but there is no such figure in the manuscript. Please correct to indicate the appropriate figure.

Response: Thanks for the sharp observation The figure referenced here is Figure 2 (now line 95). This error is now corrected.

  • Line 127: the way the sentence is written indicates that Table 1 shows predicted protein functions, but the table only shows the total number of proteins identified, not their function. Modify the text accordingly to reflect the table content. Unless the text was referring to Table 4?

Response: Thank you for this remark. The figure referenced here is indeed Table 4 (now line 132, Table 3).

  • Italicized of F. alocis in several parts of the discussion is missing. For example, lines 172, 182.

Response: We thank the reviewer for this comment. We went through the whole manuscript and italicized all F. alocis.

Reviewer 2 Report

The manuscript entitled: Proteomic characterization of the oral pathogen Filifactor alocis reveals key interactions of its RTX toxin, FtxA” reports on characterization of 10 strains (nine clinical and 1 reference strain). Both, FtxA-producing and non-producing strains were included. Large number of proteins were identified and thus this work will serve as a valuable resource for many investigators. The proteomic analysis revealed differential expression of FtxA among the strains and specific interaction with 6 proteins, including FtxB and FtxD. Overall, this work provides significant information that can be used as future leads for future experimental investigation as regards the role of FtxA in terms of contribution to virulence.  

Several comments:

  1. Line 17: oral is used twice
  2. Line 67-68: may need to revise the sentence
  3. Line 182: italicize F. alocis
  4. Line 185: meaning of abse?
  5. Gene names: italicize (e.g line 194 – ftxA)
  6. The genomic island encoding FtxA needs more information (including listing of all genes encoding members interacting with FtxA)
  7. The interactions are predicted but there is no attempt to substantiate the claims experimentally. Is there any connection between expression at a transcriptome levels under various conditions? This reviewer realizes that alocis is not easy organism to manipulate genetically
  8. The fact that the protein interactions are predicted should be included in the abstract of this paper
  9. List of the differentially-expressed proteins between FtxA-positive and FtxA-negative strains should be included in the text of the paper

Author Response

Many thanks for the time and efforts in reviewing the present manuscript and for giving these valuable and relevant comments. We have given the responses to the comments below and made changes in the manuscript accordingly and highlighted all amendments, in the revised manuscript.

Comments from Reviewer 2:

The manuscript entitled: Proteomic characterization of the oral pathogen Filifactor alocis reveals key interactions of its RTX toxin, FtxA” reports on characterization of 10 strains (nine clinical and 1 reference strain). Both, FtxA-producing and non-producing strains were included. Large number of proteins were identified and thus this work will serve as a valuable resource for many investigators. The proteomic analysis revealed differential expression of FtxA among the strains and specific interaction with 6 proteins, including FtxB and FtxD. Overall, this work provides significant information that can be used as future leads for future experimental investigation as regards the role of FtxA in terms of contribution to virulence.  

Several comments:

1. Line 17: oral is used twice

Response: We thank the reviewer for this comment. This typo is fixed now.

2. Line 67-68: may need to revise the sentence

Response: Thank you for these comments. The sentence is revised

3. Line 182: italicize F. alocis

Response: We thank the reviewer for this sharp observation. The word F. alocis (now line187) is italicized.

4. Line 185: meaning of abse?

Response: Thank you for this remark. The word is “absence”, this typo is fixed (now line 190).

5. Gene names: italicize (e.g line 194 – ftxA)

Response: The word ftxA (now 201) is italicized

6. The genomic island encoding FtxA needs more information (including listing of all genes encoding members interacting with FtxA)

Response: A more detailed description of genomic island encoding FtxA is now placed in lines 196-198.

7. The interactions are predicted but there is no attempt to substantiate the claims experimentally. Is there any connection between expression at a transcriptome levels under various conditions? This reviewer realizes that alocis is not easy organism to manipulate genetically

Response: F. alocis is still an emerging oral pathogen, for which there is no transcriptome data to support the prediction of the protein interaction. However, we think the reviewer has a valid suggestion on this point and will consider transcriptomic or genetically modification on F. alocis strains in future works. The scope of the present study was to investigate the functional conservation among the ten F. alocis strains. We hope the current context is sufficient for the standard of the current issue.

8. The fact that the protein interactions are predicted should be included in the abstract of this paper

Response: The information is now included in the abstract.

9. List of the differentially-expressed proteins between FtxA-positive and FtxA-negative strains should be included in the text of the paper

Response: A new table (table 2) with the list of the differentially expressed proteins between FtxA-positive and FtxA-negative strains was created.
